# Cytokine profiling of samples positive for *Chlamydia trachomatis* and Human papillomavirus

**Larissa Zatorre Almeida Lugo[1], Marco Antonio Moreira Puga[1]\*, Camila Mareti Bonin Jacob[2], Cacilda Tezelli Junqueira Padovani[2], Mariana Calarge Nocetti[3], Maisa Souza Tupiná[4], Ana Flávia Silva Pina[4], Jennifer Naed Martins de Freitas[1], Alda Maria Teixeira Ferreira[2], Carlos Eurico dos Santos Fernandes[2], Adriane Cristina Bovo[5], Júlio César Possati Resende[6], Inês Aparecida Tozetti[1,2]**

1 Postgraduate Program of Infectious and Parasitary Diseases from Medicine School, Federal University of Mato Grosso do Sul, Campo Grande, Mato Grosso do Sul, Brazil, 2 Bioscience Institute from the Federal University of Mato Grosso do Sul, Campo Grande, Mato Grosso do Sul, Brazil, 3 Faculdade de Ciências Farmacêuticas, Nutrição e Alimentos, UFMS, Campo Grande, Mato Grosso do Sul, Brazil, 4 Medicine School, Federal University of Mato Grosso do Sul, Campo Grande, Mato Grosso do Sul, Brazil, 5 Department of Cancer Prevention, Barretos Cancer Hospital, Campo Grande, MS, Brazil, 6 Department of Cancer Prevention, Barretos Cancer Hospital, Barretos, São Paulo, Brazil

\* marco.m.puga@gmail.com

**Data Availability Statement:** All relevant data are within the manuscript.

**Funding:** This work was supported by the Federal University of Mato Grosso do Sul Foundation -

## Abstract

Persistent human papillomavirus (HPV) infection is closely associated with cervical carcinoma. Co-infection in the endocervical environment with other microorganisms, such as *Chlamydia trachomatis*, may increase the risk of HPV infection and neoplastic progression. While in some individuals, *Chlamydia trachomatis* infection is resolved with the activation of Th1/IFN-γ-mediated immune response, others develop a chronic infection marked by Th2-mediated immune response, resulting in intracellular persistence of the bacterium and increasing the risk of HPV infection. This work aimed to quantify cytokines of the Th1/Th2/Th17 profile in exfoliated cervix cells (ECC) and peripheral blood (PB) of patients positive for *Chlamydia trachomatis* DNA, patients positive for Papillomavirus DNA, and healthy patients. Cytokine levels were quantified by flow cytometry in ECC and PB samples from patients positive for *C. trachomatis* DNA (n = 18), patients positive for HPV DNA (n = 30), and healthy patients (n = 17) treated at the Hospital de Amor, Campo Grande-MS. After analysis, a higher concentration of IL-17, IL-6, and IL-4 (p <0.05) in ECC; INF-γ and IL-10 (p <0.05) in PB was found in samples from patients positive for *C. trachomatis* DNA compared to samples from healthy patients. When comparing samples from patients positive for HPV DNA, there was a higher concentration of cytokines IL-17, IL-10, IL-6, and IL-4 (p <0.05) in ECC and IL-4 and IL-2 (p <0.05) in PB of patients positive for *C. trachomatis* DNA. These results suggest that induction of Th2- and Th17 mediated immune response occurs in patients positive for *C. trachomatis* DNA, indicating chronic infection. Our results also demonstrate a high concentration of pro-inflammatory cytokines in ECC of patients positive for *C. trachomatis* DNA.

UFMS/MEC - Brazil, the National Council for Scientific and Technological Development (CNPq) (Grant number: 406148/2016-3) and the Coordination for the Improvement of Higher Education Personnel (CAPES). This study was financed in part by the Coordenação de Aperfeiçoamento de Pessoal de Nível Superior - Brasil (CAPES) - Finance Code 001. The funders had no role in study design, data collection and analysis, decision to publish, or preparation of the manuscript.

**Competing interests:** The authors have declared that no competing interests exist.

## Introduction

Maintaining a healthy vaginal microbiota is crucial for protection against Sexually Transmitted Infections (STIs), such as those caused by *Chlamydia trachomatis* (*C. trachomatis*) and Human Papillomavirus (HPV) [1]. HPV infection is closely associated with the development of cervical cancer by co-infection with *C. trachomatis*, which induces a deviation of the immune response to the Th2 profile, favoring the viral infection [2, 3].

Bacterial infection by *C. trachomatis* is the most common STI in the world, surpassing gonococcal and treponemal infection [4], and poses a major public health concern. Every year, approximately 131 million new cases of genital infection caused by *C. trachomatis* are diagnosed worldwide, of which 70% are asymptomatic [5]. In individuals who acquire bacterial infection before the age of twenty, recurrence and persistence of the bacterium are observed, and the immunity developed is partially protective, especially against the different serotypes. Successive episodes of infection increase the risk of sequelae and susceptibility to co-infection [6–8].

*C. trachomatis* shows tropism for cervical columnar epithelium, and although infection usually begins in the endocervix, the bacterium can be found at other sites, such as the vaginal canal and fallopian tube [6, 9]. The development cycle is biphasic, with the coexistence of two distinct morphological forms, the elementary body (EB) and the reticulate body (RB) [6].

After infection, antigen-presenting cells (APC) recognize the bacteria at the site of infection and initiate the release of pro-inflammatory cytokines, such as IFN-γ and IL-12. These cytokines, in turn, activate Natural Killer (NK) cells and induce the activation of CD8+ and CD4 + T-cells. The CD4+ effector T-lymphocytes, through the production of cytokines, help activate different immune response profiles. The expected immune response profile to *C. trachomatis* is mediated by Th1 lymphocytes, responsible for activating B lymphocytes and their subsequent differentiation into plasma cells, secreting specific antibodies against *C. trachomatis*. However, in some cases, due to the biphasic bacterial cycle, deviation in the immunological response towards the Th2 profile (increased levels of IL-4 and IL-6), which correlates to a poor infection prognosis, may occur [7].

A co-infection with other microorganisms, such as HPV, can facilitate the deviation to Th2-type immune response, which may result in the potentiation of a viral and bacterial infection or even an HPV-dependent progression to carcinogenesis. Co-infection with *C. trachomatis* and HPV may lead to increased expression of Ki-67 (a marker for cell proliferation) and E6 and E7 proteins (HPV oncoproteins), in addition to immune modulation by inhibiting apoptosis [8]. In relation to HPV infection, a predominance of Th1 polarization is expected, with the production of IFN-γ which is important for viral elimination and control of neoplastic progression [10, 11]. Polarization of the immune response towards Th2-type (IL-4, IL-5, IL-10, and IL-13) favors viral persistence and in cases of co-infection, bacterial elimination is compromised [12, 13].

During the last few years, the role of the Th17 response profile in the resolution of HPV infections and *C. trachomatis* has been investigated [14–16]. Studies show that mice infected with *C. trachomatis* that do not produce IL-17 generate an immunological response predominated by the Th1/ IFN-γ pathway. However, those deficient in the production of IFN-γ and infected by this same bacterium display higher levels of IL-17 [17]. Given that the change in cytokine profile may be associated with the prevalence of bacterial infection and a poorer prognosis, the present study aimed to quantify the levels of cytokines IL-2, IL-4, IL-6, IL-10, IL-17A, INF- γ, and TNF in peripheral blood and exfoliated cervical cell (ECCs) of patients positive for *C. trachomatis* DNA and compare them with patients positive and negative for HPV DNA. In this way, we will be able to understand more about the interaction between the

immune response and these two microorganisms, if the cervical or systemic microenvironment can somehow influence viral persistence and if the presence of C. Trachomatis can modulate this environment.

## Materials and methods

### Study population and sample collection

A case-control, cross-sectional study was carried out. The samples used in the study were obtained from 493 female patients, aged over 18 years, who reported at the Hospital de Amor (Fundação PIO XII), Campo Grande unit—MS, for cytological examination in July 2016. All patients who agreed to participate in this study signed a written Informed Consent Form. The study was approved by the Research Ethics Committee of the Federal University of Mato Grosso do Sul and the Fundação PIO XII under protocol no. 1468457 and 1635895, respectively.

The case-control study groups comprised samples of exfoliated cervix cells (ECC) and peripheral blood (PB) from eighteen patients positive for *Chlamydia trachomatis* DNA, thirty HPV DNA-positive patients, and eighteen healthy patients (*Chlamydia trachomatis* and HPV DNA-negative*)* selected within the universe of 493 samples. The mean ages were 46.2 years for *C. trachomatis* DNA-positive patients, 44.7 years for those positive for HPV DNA, and 43.5 years for the healthy group (control group). All groups were negative for anti-HIV and anti-Treponema pallidum antibodies and hepatitis B surface circulating antigen (HBsAg) to exclude systemic and cervical co-infections that could influence cytokine levels.

ECC and PB samples were selected in a non-probabilistic manner for convenience. The collection of ECC samples was performed before the collection of the cytology material, according to the literature [18, 19]. Samples were obtained and processed according to Bonin et al. (2019) [20].

Serum samples for cytokine evaluation and identification of anti-HIV anti-*Treponema pallidum* antibodies and HBsAg were collected by venous puncture. These samples were processed to obtain the serum, then divided into aliquots and stored at -80˚C until further processing.

All 493 samples were submitted to the following sequence of tests.

### Detection of human papillomavirus and β-globin

PCR was performed using the samples to detect HPV DNA with primers PGMY 09/11 [21] and β-globin with primers PC04/GH20 [22]. HPV-positive samples were subjected to viral genotyping by the type-specific PCR (TS-PCR) technique to determine high-risk HPV (HR-HPV) types i.e. HPV-16, 18, 31, 33, and 45, and low-risk HPV (LR-HPV) types i.e. HPV-6 and 11. Samples that could not have their viral type determined by TS-PCR were subjected to Restriction Fragment Length Polymorphism (RFLP) with enzymes *Dde*I, *Hae*III, *Rsa*I, and *Pst*I. The restriction fragments were visualized in 1.5% and 3.0% agarose gel with ethidium bromide staining to visualize DNA under ultraviolet light. Molecular weights were determined by comparing with a 100 bp DNA ladder. Finally, Sanger sequencing was carried out for samples that could not be genotyped by other techniques, using the equipment ABI-Prism 3500 Genetic Analyzer (Applied Biosystems, Foster City, California, USA), with the entire protocol following Bonin et al. (2019) [20].

### Detection of *Chlamydia trachomatis*

For identification of *C. trachomatis*, all samples positive for amplification of the β-globin gene were subjected to PCR amplification of the 241 bp region of the bacterial genome. Following

the methodology of Santos et al. (2003) [23], the primers KL1-5' TCCGGAGCGAGTTAC-GAAGA 3' and KL2-5' AATCAATGCCCGGGATTGGT 3' were used.

To 45 μL of PCR solution prepared from each sample, 5 μL of 50 ng/μL DNA was added. The reaction was processed in a thermocycler programmed for 94°C for 5 minutes for denaturation and then for 25 cycles of 94°C for 30 seconds, 55°C for 30 seconds, and 72°C for 30 seconds. After 25 cycles, a final extension at 72°C was carried out for 7 minutes. The PCR products were visualized in 1.5% and 3.0% agarose gel with ethidium bromide staining to visualize DNA under ultraviolet light. Molecular weights were determined by comparing with a 100 bp DNA ladder.

## Detection of anti-HIV1 and 2, anti-*Treponema pallidum* and HBsAg antibodies

The determination of infections caused by human immunodeficiency virus (HIV), hepatitis B (HBV), and *Treponema pallidum*, the causative agent of syphilis, was carried out by the following rapid tests: HIV Test Bioeasy (Standard Diagnostics, Inc., South Korea), Vikia® HBsAg (Biomerieux, France), and TR DPP Sífilis DUO Bio Manguinhos (Oswaldo Cruz Foundation, Brazil), respectively. All three tests are qualitative, based on immunochromatographic techniques, and approved by the National Health Surveillance Agency (ANVISA) responsible for regulating, controlling, and supervising products and services involving risks to public health in Brazil.

## Composition of groups for the case-control study

The groups were divided in ECC (n = 18) and PB samples (n = 16) from C. trachomatis DNA-positive patients (n = 18), HPV DNA-positive patients (n = 30), and healthy patients (n = 17) (Chlamydia trachomatis and HPV-DNA negative). All samples were negative for anti-HIV1 and 2, anti-Treponema pallidum, and HBsAg.

## Cytokine determination

Cytokine levels were measured in PB and ECC samples of the three groups analyzed: C. trachomatis-positive, HPV-positive, and healthy patients. The cytokines IL-2, IL-4, IL-10, IL-17A, INF-γ, and TNF were quantified using the human Cytokine Bead Array (CBA) Th1/Th2/Th17 (BD Biosciences, San Jose, CA, USA). The preparation of standards, beads, samples, and protocols for configuring the flow cytometer and data acquisition were performed according to the manufacturer's instructions. The samples were acquired using a FACS Canto II flow cytometer (BD Biosciences, San Jose, CA, USA) and analyzed using the FCAP Array V. 3.0 software (Becton Dickinson, Franklin Lakes, NJ, USA). The results were based on standard concentration curves and expressed in pg/mL.

## Statistical analysis

The participants were grouped into positive for *C. trachomatis*, positive for HPV, and healthy patients. The cytokine levels were expressed as Mean ± standard error (SEM). The Mann-Whitney U test was used to analyze the differences between the HPV group vs. *C. trachomatis group* and *C. trachomatis group* vs. healthy patients. Analysis of variance and T test were used between the groups. The results were considered significant when $p < 0.05$. The figures were prepared using the GraphPad Prism 5.0 program (Graph-Pad Software, San Diego, CA, USA).

**Table 1. Socioepidemiological characteristics of patients treated at the Hospital de Amor, Campo Grande, MS, Brazil.** *Chlamydia trachomatis* DNA-positive patients (n = 18), HPV DNA positive patients (n = 30), and healthy patients (n = 17).

| Variables | *C. trachomatis* | | HPV | | No infections | |
|---|---|---|---|---|---|---|
| | No. | (%) | No. | (%) | No. | (%) |
| Number of patients | 18 | | 30 | | 17 | |
| Average age (range) | 43.2 (26–62) | | 44.7 (26–64) | | 43.9 (26–64) | |
| Average First sexual intercourse | 16.9 | | 17.9 | | 18.9 | |
| Condom use | | | | | | |
| Yes | 3 | (16.7) | 8 | (26.6) | 3 | (14.6) |
| No | 15 | (83.3) | 22 | (73.4) | 14 | (82.3) |
| Preventive | | | | | | |
| More than once a year | 2 | (11.1) | 2 | (6.7) | 3 | (17.6) |
| Once a year | 14 | (77.8) | 20 | (66.7) | 9 | (52.9) |
| Less than once a year | 2 | (11.1) | 5 | (16.7) | 5 | (29.5) |
| Missing | - | - | 3 | (10.0) | - | - |

## Results

### Socioepidemiological characteristics

The socioepidemiological characterization of the participants in this study is shown in Table 1. Due to the low number of co-infection samples, it was impossible to include the HPV x *Chlamydia trachomatis* co-infection in analyses (3/25) (S1 Table).

### Viral genotyping

Among the HPV-positive samples, the most frequent genotypes were HPV 6/11 (35.5%), HPV 59 (28.8%), and HPV 45 (17.7%); 42.2% of the samples had multiple infections (more than one type of HPV). Fig 1 shows the genotyped HPV and infection by more than one viral type in 40.0% (12/30) samples (Fig 1B). In this figure the viral types are classified as high oncogenic risk (HR-HPV), low oncogenic risk (LR-HPV), and possible high oncogenic risk (found in high and low-risk lesions) [24].

### Comparison of cytokine levels in samples positive for C. trachomatis DNA with those in HPV DNA-positive samples

The levels of cytokines INF-γ, TNF, IL-17A, IL-10, IL-6, IL-4, and IL-2 (pg/mL) were measured in PB and ECC samples of 18 *C. trachomatis* DNA-positive patients. One serum sample was insufficient for analysis, leaving 17 serum samples (mean age 46.2 years). The levels of the same cytokines were measured in PB and ECC of 18 healthy patients (mean age 43.6 years).

Levels of INF-γ (p = 0.004), IL-10 (p = 0.025), and IL-2 (p = 0.009) in PB samples of *C. trachomatis* DNA-positive patients were higher (n = 16) as compared to that in PB samples from healthy patients (n = 18). As for the cytokines detected in EECs, there was a higher level of IL-17 (p = 0.048), IL-6 (p = 0.019), and IL-4 (p = 0.008) in the *C. trachomatis* DNA-positive samples (n = 18) as compared to healthy patient samples (n = 18) (Table 2).

### Cytokine levels in samples positive for *C. trachomatis* DNA and in HPV DNA-positive samples

Levels of cytokines INF-γ, TNF, IL-17A, IL-10, IL-6, IL-4e, and IL-2 (pg/mL) were measured in PB and ECC samples of 18 *C. trachomatis* DNA-positive patients (mean age 43.6 years) and

a.

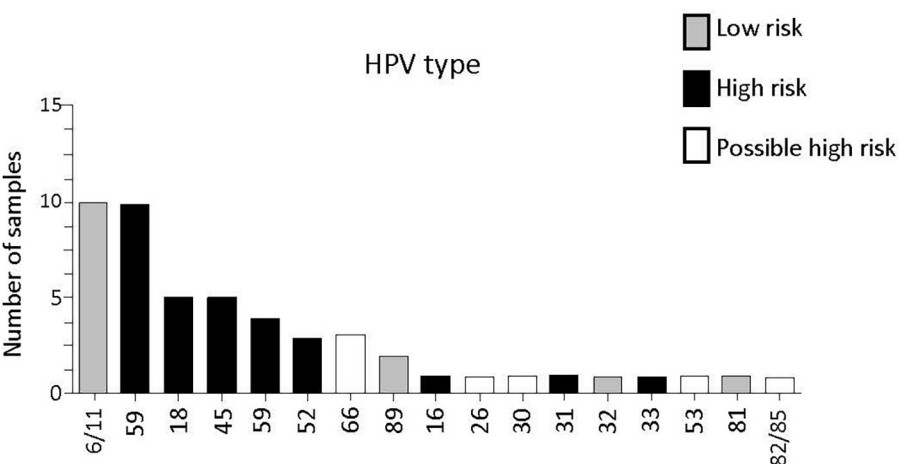

b.

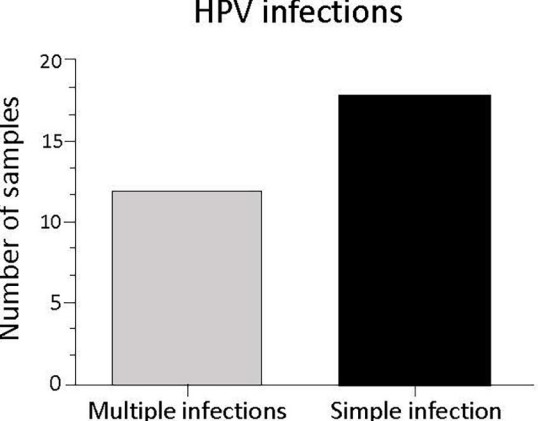

**Fig 1. Viral genotyping and presence of multiple (by more than one viral type) or simple infections (by a single viral type) of samples from patients treated at Hospital de Amor, Campo Grande, MS, Brazil.** (a.) Viral HPV types detected in samples with multiple or simple infections (n = 30) and (b.) Frequency of simple and multiple HPV infection (n = 30).

30 HPV DNA-positive patients (mean age 44.7 years). In the ECC samples, there was a significant increase in the concentrations of IL-17 (p = 0.011), IL-10 (p = 0.027), IL-6 (p = 0.000), and IL-4 (p = 0.002) among *C. trachomatis* DNA-positive samples when compared to those positive for HPV DNA. However, there was no significant difference in the levels of IFN-γ, TNF, and IL-2 (p > 0.05) between the two groups (Fig 2).

When PB samples were assessed, higher levels of IL-4 (p = 0.020) and IL-2 (p = 0.001) were observed among *C. trachomatis* DNA-positive samples as compared to HPV DNA-positive samples. There was no significant difference in the levels of the other cytokines (p > 0.05) (Table 3).

**Table 2. Cytokine levels in the exfoliated cervical cell (ECC) and peripheral blood (PB) of patients with *Chlamydia trachomatis* infections and group control (no infections).**

| Sample | Cytokines | *C. tracomatis* positive | *Healthy patients* | Value of p |
|---|---|---|---|---|
| | | (Serum n = 17 ECC n = 18) | (n = 18) | |
| | | Mean [pg/mL] ± SEM | Mean [pg/mL] ± SEM | |
| PB | IFN-γ | 4.60±0.21 | 3.7±0.21 | 0.006 |
| | TNF | 5.34±0.44 | 4.54±0.45 | 0.205 |
| | IL-17 | 33.45±4.45 | 24.45±4.78 | 0.186 |
| | IL-10 | 4.73±0.07 | 4.50±0.07 | 0.030 |
| | IL-6 | 3.77±0.39 | 3.59±0.40 | 0.752 |
| | IL-4 | 10.57±3.78 | 5.06±3.89 | 0.318 |
| | IL-2 | 23.23±9.77 | 2.60±10.05 | 0.151 |
| ECC | IFN-γ | 3.94±0.12 | 3.75±0.12 | 0.287 |
| | TNF | 5.79±0.88 | 4.57±0.88 | 0.334 |
| | IL-17 | 30.00±3.27 | 20.49±3.27 | 0.048 |
| | IL-10 | 5.51±0.43 | 4.58±0.42 | 0.129 |
| | IL-6 | 200.16±42.88 | 51.39±42.88 | 0.019 |
| | IL-4 | 5.61±0.17 | 4.93±0.17 | 0.008 |
| | IL-2 | 2.60±0.06 | 2.76±0.06 | 0.078 |

Note: The results are expressed as mean ± SEM

Mann-Whitney U-test was performed and p <0.05 was considered statistically significant

ECC—exfoliated cervical cells.

## Discussion

Persistence of certain infectious microbes such as *C. trachomatis* in the endocervical epithelium may be associated with neoplastic lesions and infertility [4]. The constant recurrence of *C. trachomatis* infection is due to change in modulation of the immune response, compromising the response mediated by CD8+ T cells. Any bacterial infection initially elicits an inflammatory response in the host with subsequent activation of the innate immune response, and this response is self-limiting. However, the absence of adequate immune modulation results in chronic inflammation which causes large-scale tissue damage resulting in a site conducive for the entry of new pathogens [7].

HPV is one such pathogen that benefits from lesions in the epithelium and the microenvironment produced during infection by *C. trachomatis*. The association between HPV and *C. trachomatis* infection enhances neoplastic transformation, and this may be largely based on the modulation of the immune response. Studies show that infection by *C. trachomatis* stimulates a Th1-type cellular immune response, which facilitates infection clearance. However, a humoral response (Th2-type) is also observed in a few cases of *C. trachomatis* infection, which is associated with persistent infection by this bacterium. This modulation of the immune response also favors HPV infection and persistence [17, 25–28]. Additionally, it is also noteworthy that HPV infection modulates the immunological response towards a predominance of Th2-type response (IL-4, IL-5, IL-10, and IL-13), which favors persistence of bacterial infection [12, 29].

Presence of the bacterium *C. trachomatis* in the cervical microenvironment simultaneously with HPV may aggravate the overall infection. It has been reported that the induction of an inflammatory environment in response to the bacterial infection aggravates HPV infection and increases the chances of neoplastic alteration [25, 28]. Although there are reports

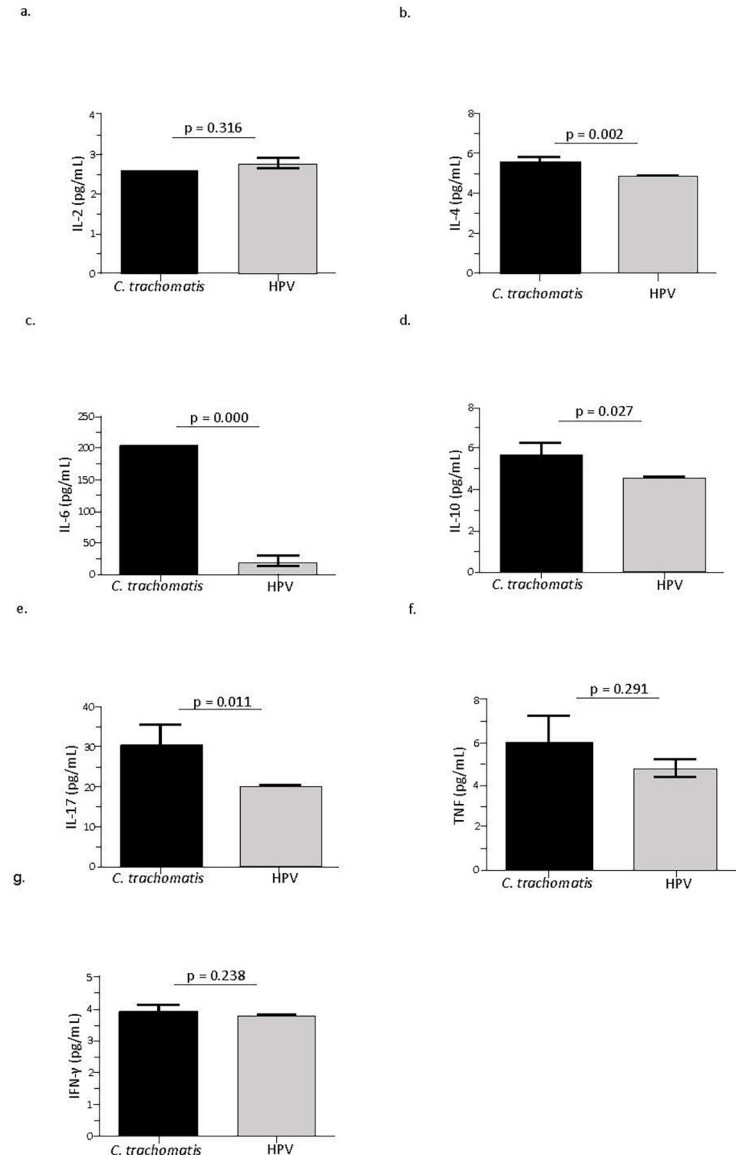

**Fig 2. Cytokine levels (pg/ml) in exfoliated cervical cells (ECC) from *Chlamydia trachomatis* (*C. trachomatis*) DNA-positive samples and human papillomavirus (HPV) DNA-positive samples.**

indicating that co-infections may increase the chances of progression of HPV infection, there is no way to know which microorganism was the first to infect the epithelium.

The ideal immune response profile for bacterial elimination would be marked by a predominance of Th1 cytokines (IL-12 / IFN-γ) which are responsible for the activation of cellular immunity for elimination of infected cells through the activation of phagocytic cells [33]. However, the presence of pro-inflammatory cytokines, such as IL-6, is related to a higher risk of complications from this infection. Studies show that host cells infected with *C. trachomatis* release this pro-inflammatory cytokine throughout the growth cycle of *C. trachomatis* which, along with the release of TNF and IL-10, leads to a poorer prognosis of the infection [30].

In the present study, there was a predominance of Th2 and Th17 response profiles in the samples from patients positive for *C. trachomati*s DNA. In the ECC samples from patients

**Table 3. Cytokine levels in the peripheral blood (PB) samples of patients with *Chlamydia trachomatis* infections and HPV infection.**

| Sample | Cytokines | *C. trachomatis* positive (n = 17) Mean [pg/mL] ± SEM | HPV positive (n = 30) Mean [pg/mL] ± SEM | Value of p |
|---|---|---|---|---|
| PB | IFN-γ | 4.66±0.32 | 4.23±0.23 | 0.292 |
|  | TNF | 5.23±0.46 | 4.72±0.33 | 0.378 |
|  | IL-17 | 35.28±7.17 | 43.39±5.07 | 0.361 |
|  | IL-10 | 4.69±0.12 | 4.78±0.08 | 0.555 |
|  | IL-6 | 3.95±1.76 | 5.43±1.24 | 0.498 |
|  | IL-4 | 5.36±0.12 | 4.98±0.08 | 0.010 |
|  | IL-2 | 9.00±1.46 | 2.64±1.03 | 0.001 |

Note: The results are expressed as mean ± SEM

Mann-Whitney U-test was performed and p <0.05 was considered statistically significant.

positive for *C. trachomati*s DNA, it was possible to observe predominance of the pro-inflammatory cytokines IL-6 and IL-17 in addition to cytokines predominant in the Th2 response (IL-10 and IL-4) (Fig 2). Studies show that IL-10 plays an important role in regulating immune response to infectious agents by suppressing the secretion of pro-inflammatory cytokines and thereby impairing the elimination of pathogens, leading to persistence and worsening of the infection [31]. Monocytes (CD14+ cells) were found to be the main source of IL-10 in individuals with *C. trachomatis* compared to samples from uninfected, healthy individuals [32]. The mechanisms that regulate IL-10 secretion in *C. trachomatis* are not completely understood. However, it has been reported that exogenous IL-10 can inhibit the secretion of IL-6, IL-8, and TNF- α in cell line [33]. In the present study, this suppression or alteration in the response profile could have been ineffective since there was a predominance of pro-inflammatory cytokines, such as IL-6, in the infection-induced microenvironment.

The predominance of IL-4 and IL-2 was also observed in the ECC and PB samples from patients positive for *C. trachomati*s DNA, and these cytokines are predominant in the Th2 response. These results corroborate those found by Vicetti et al. (2013) [34], who demonstrated that peripheral blood mononuclear cells (PBMC) of women infected with *C. trachomatis* were polarized towards a Th2 response characterized by the production of IL-4 and IL-2.

For bacterial infection caused by *C. trachomatis*, in addition to the Th1 response, Th17 cells are also associated with reinfection and epithelial injury [16]. In the present study, the concentration of IL-17 was found to be higher in the samples from patients positive for *C. trachomati*s DNA as compared to samples from healthy patients and HPV DNA-positive patients. The presence of Th17 cells has been associated with some autoimmune diseases and diseases caused by extracellular pathogens, and this response is known to cause potential tissue damage [27]. Studies show an increase in IL-17 levels in the genital tract of women infected with *C. trachomatis* and gonococci, suggesting that IL-17 production may be induced in response to these pathogens. Some studies also demonstrated that Th17 cells are necessary for host defense against infection by these bacterium [35–38].

Th17 cells are considered potent inducers of autoimmunity by promoting tissue inflammation and activation of the innate immune system [37, 39, 40]. However, in case of HPV co-infection or a worsening of bacterial infection, these inflammatory mediators secreted by Th17 cells may contribute to neoplastic progression by positive regulation of immune suppressive cells of the adaptive and innate immune system as well as the induction of the inflammatory reaction at the site of infection [16, 39].

Differentiation of CD4+ T cells into Th17 subtype is dependent on the expression of the transcription factor RORγt, which is induced by IL-6 and IL-23 cytokines. Thus, the presence of pro-inflammatory IL-6 at the site of infection helps in the differentiation of Th17 cells. This cytokine also induces tissue damage and may facilitate infection by other pathogens due to micro lesions development [40, 41]. The expression level of IL-6 shows a positive correlation with the proportion of Th-17 cells [15]. In the present study, IL-6 levels were also higher in ECC samples from patients positive for *C. trachomatis* DNA compared to samples from healthy and HPV DNA-positive patients.

IL-6 production is required for an optimal initial host immune response against the bacterium, but a sustained elevated level induces local tissue damage [15, 42]. IL-6 has also been found to be elevated in patients with adverse obstetric outcomes, such as recurrent abortion. There is compelling evidence indicating that altered systemic trans-signaling of IL-6 occurs in women prone to recurrent abortion, with excessive bioavailability of IL-6 potentially inhibiting CD4+ T-regulatory cell generation [15, 41].

Increase in the production of pro-inflammatory cytokines such as IL-6 and IL-17 induced by *C. trachomatis* infection has shown that chronic inflammatory environment facilitates neoplastic progression by stimulating cell proliferation and secretion of pro-angiogenic and immunosuppressive factors, and thereby contributing both to the onset of HPV infection as well neoplastic progression [25, 28, 41–43].

Pro-inflammatory cytokines can act in conjunction with interferon-γ (IFN-γ), aiding in the activation of phagocytic cells to limit infection. IFN-γ is produced when the elementary bodies (EBs) infect host cells resulting in activation of the immune response which leads to growth inhibition and replication blockage of reticulate bodies (RBs). When *C. trachomatis* is exposed to IFN-γ, the bacteria stop replicating and remain in latent form, and when the production of IFN-γ decreases, the RBs begin to replicate again, continuing the cycle and reactivating the infection [7, 41].

In the present study, there was no predominance of IFN-γ at the site of infection indicating bacterial replication. However, upon considering the systemic response, there was a predominance of IL-10 and IFN-γ among the PB samples from patients positive for *C. trachomatis* DNA compared to those from healthy patients. This can be justified by a period of development of the infection or by the presence of other pathogens not identified systemically since IFN-γ, an important participant of the Th1 response, plays a crucial role in host defense against intracellular infection by stimulating cellular immunity to eliminate bacteria and infected cells.

IL-10 was also predominant in serum of *C. trachomatis* DNA-positive patients. Studies show that this cytokine can act as a protective factor at the systemic level. As reported in other studies, IL-10 was produced at high concentrations in PBMCs of women with normal pregnancy compared to women with a history of unexplained recurrent abortion [44, 45]. Therefore, IL-10 has emerged as an important Th2 cytokine since it is directly involved in regulating Th1 activity, thus playing an essential immunoregulatory role.

Regarding the immune response to *C. trachomatis*, a predominance of Th1 profile responsible for the activation of phagocytic cells is expected. However, the presence of pro-inflammatory cytokines, such as IL-6, is associated with an increased risk of complications from this infection. Studies show that host cells infected with *C. trachomatis* release pro-inflammatory cytokines throughout their growth cycle, which together with IL-1b, TNF, and IL-10, are associated with poorer prognosis of the infection [46, 47].

This study is not definitive or conclusive on the subject. Still, it contributes to understanding more about the interaction between the immune response and these two microorganisms, if the cervical or systemic microenvironment can somehow influence viral persistence and if

the presence of C. trachomatis can modulate this environment. The work aimed to consider the cytokines produced by the different groups with samples collected from the cervix and blood. However, studies by our group associated immunophenotyping with cytokine dosage to characterize better the patterns obtained. One of these studies is already published [48].

In the present study, we chose to compare the cytokines involved in the immune response to HPV and *C. trachomatis*, demonstrating the possible potentiating effect that cytokines involved in the Th2 and Th17 response could have on co-infections. However, due to the limited sample size and not being a cohort study, it was not possible to assess the cytokine profile in HPV and *Chlamydia trachomatis* co-infection and the evolution of the cytokine profile during the infectious process.

## Conclusion

There was a predominance of pro-inflammatory cytokines in the samples from patients positive for *C. trachomatis* DNA, as well as a predominance of Th2 and Th17 cytokine profiles, which could be correlated to poorer prognosis of the bacterial infection. It is emphasized that the local inflammatory processes can cause damage to the epithelium and facilitate entry of other pathogens, for example, infection by HPV, viral persistence, and subsequent neoplastic progression.

## Supporting information

**S1 Table. Cytokines concentrations from coinfected individuals HPV/Chlamydia.**
(PDF)

## Acknowledgments

Federal University of Mato Grosso do Sul (UFMS), Ministry of Education and Science (MEC), and the Hospital de Amor, Campo Grande-MS unit.

## Author Contributions

**Conceptualization:** Larissa Zatorre Almeida Lugo, Cacilda Tezelli Junqueira Padovani, Alda Maria Teixeira Ferreira, Carlos Eurico dos Santos Fernandes, Inês Aparecida Tozetti.

**Data curation:** Larissa Zatorre Almeida Lugo, Marco Antonio Moreira Puga, Camila Mareti Bonin Jacob, Cacilda Tezelli Junqueira Padovani, Alda Maria Teixeira Ferreira, Carlos Eurico dos Santos Fernandes, Adriane Cristina Bovo, Júlio César Possati Resende, Inês Aparecida Tozetti.

**Formal analysis:** Camila Mareti Bonin Jacob, Carlos Eurico dos Santos Fernandes.

**Funding acquisition:** Inês Aparecida Tozetti.

**Investigation:** Marco Antonio Moreira Puga, Camila Mareti Bonin Jacob, Cacilda Tezelli Junqueira Padovani, Mariana Calarge Nocetti, Maisa Souza Tupiná, Ana Flávia Silva Pina, Jennifer Naed Martins de Freitas, Alda Maria Teixeira Ferreira, Adriane Cristina Bovo, Júlio César Possati Resende, Inês Aparecida Tozetti.

**Methodology:** Larissa Zatorre Almeida Lugo, Marco Antonio Moreira Puga, Camila Mareti Bonin Jacob, Cacilda Tezelli Junqueira Padovani, Mariana Calarge Nocetti, Maisa Souza Tupiná, Ana Flávia Silva Pina, Jennifer Naed Martins de Freitas, Alda Maria Teixeira Ferreira, Adriane Cristina Bovo, Júlio César Possati Resende, Inês Aparecida Tozetti.

**Project administration:** Alda Maria Teixeira Ferreira, Inês Aparecida Tozetti.

**Resources:** Inês Aparecida Tozetti.

**Supervision:** Marco Antonio Moreira Puga, Alda Maria Teixeira Ferreira, Carlos Eurico dos Santos Fernandes, Inês Aparecida Tozetti.

**Validation:** Inês Aparecida Tozetti.

**Visualization:** Camila Mareti Bonin Jacob, Inês Aparecida Tozetti.

**Writing – original draft:** Larissa Zatorre Almeida Lugo, Marco Antonio Moreira Puga, Camila Mareti Bonin Jacob, Cacilda Tezelli Junqueira Padovani, Mariana Calarge Nocetti, Maisa Souza Tupiná, Ana Flávia Silva Pina, Jennifer Naed Martins de Freitas, Alda Maria Teixeira Ferreira, Carlos Eurico dos Santos Fernandes, Adriane Cristina Bovo, Júlio César Possati Resende, Inês Aparecida Tozetti.

**Writing – review & editing:** Larissa Zatorre Almeida Lugo, Marco Antonio Moreira Puga, Camila Mareti Bonin Jacob, Cacilda Tezelli Junqueira Padovani, Mariana Calarge Nocetti, Maisa Souza Tupiná, Ana Flávia Silva Pina, Jennifer Naed Martins de Freitas, Alda Maria Teixeira Ferreira, Carlos Eurico dos Santos Fernandes, Adriane Cristina Bovo, Júlio César Possati Resende, Inês Aparecida Tozetti.

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
