## [Decision Letter · Decision Letter 0]

26 Jul 2022

PONE-D-22-04538

Cytokine profiling of samples positive for Chlamydia trachomatis and Human papillomavirus

PLOS ONE

Dear Dr. Puga,

Thank you for submitting your manuscript to PLOS ONE. After careful consideration, we feel that it has merit but does not fully meet PLOS ONE’s publication criteria as it currently stands. Therefore, we invite you to submit a revised version of the manuscript that addresses the points raised during the review process.

We look forward to receiving your revised manuscript.

Kind regards,

Thomas Forsthuber

Academic Editor

PLOS ONE

a) Did participants provide their written or verbal informed consent to participate in this study?

“This work was supported by the Federal University of Mato Grosso do Sul Foundation - UFMS/MEC - Brazil, the National Council for Scientific and Technological Development (CNPq) (Grant number: 406148/2016-3) and the Coordination for the Improvement of Higher Education Personnel (CAPES). This study was financed in part by the Coordenação de Aperfeiçoamento de Pessoal de Nível Superior - Brasil (CAPES) - Finance Code 001”

“Federal University of Mato Grosso do Sul (UFMS), Ministry of Education and Science (MEC), and the Hospital de Amor, Campo Grande-MS unit.”

“This work was supported by the Federal University of Mato Grosso do Sul Foundation - UFMS/MEC - Brazil, the National Council for Scientific and Technological Development (CNPq) (Grant number: 406148/2016-3) and the Coordination for the Improvement of Higher Education Personnel (CAPES). This study was financed in part by the Coordenação de Aperfeiçoamento de Pessoal de Nível Superior - Brasil (CAPES) - Finance Code 001”

Reviewers' comments:

Reviewer's Responses to Questions

**Comments to the Author**

1. Is the manuscript technically sound, and do the data support the conclusions?

Reviewer #1: Partly

Reviewer #2: No

2. Has the statistical analysis been performed appropriately and rigorously? 

Reviewer #1: Yes

Reviewer #2: Yes

3. Have the authors made all data underlying the findings in their manuscript fully available?

Reviewer #1: Yes

Reviewer #2: Yes

4. Is the manuscript presented in an intelligible fashion and written in standard English?

Reviewer #1: Yes

Reviewer #2: No

5. Review Comments to the Author

Reviewer #1: In this manuscript, the authors described the quantification of profile of Th1/Th2/Th17 cytokine in exfoliated cervix cells (ECC) and in peripheral blood (PB) of patients positive for Chlamydia trachomatis (Ct) DNA, patients positive for Papillomavirus DNA, and healthy patients. The results replicate other previous published works with some new data to the field. The paper has provided useful information on immune response to important STI pathogens, Ct and HPV. However, a reduced number of samples, the lack of details in patient and sample detections, and the limited interpretations on the results make it hard to draw the strong conclusions. In addition, some reference were displaced or improper. Some comments to improve the quality of this study are as below.

Major Concerns：

Methods:

1. Patients: What is the age range of the individuals?

2. What is the rationale behind the PCR testing HPV together with the human β-globin gene?

3. Lines 166 and 538: a wrong reference (23) for Ct primers and detection.

Results:

1. If could be nice to include the cytokine levels from the individuals that had co-infection of Ct and HPV.

2. Lines 267-268: the tittle was unclear here. Perhaps it should be “comparison of cytokine levels in samples positive for C. trachomatis DNA with those in HPV DNA-positive samples”.

3. Table 2: column 4 “C. tracomatis negative” should be healthy control or no infections?

4. If could be interesting to compare whether the cytokine levels with the individuals that had HPV DNA were different from the healthy controls.

5. Figures 1 and 2 were blurred.

Minor：

1. Lines 77-78: “the elementary body (EB) that is extracellular, metabolically inactive…” this is not true. Recent evidence suggest that EB has low metabolic ability. Omsland A, Sixt BS, Horn M, Hackstadt T.FEMS Microbiol Rev. 2014 Jul;38(4):779-801. doi: 10.1111/1574-6976.12059. Epub 2014 Feb 24.

2. Line 92: use a reference with women study is better.

Reviewer #2: Lugo et al, in the manuscript "Cytokine profiling of samples positive for Chlamydia trachomatis and Human

papillomavirus" profile clinical samples from cohorts of about 500 patients attending an outpatient department in a hospital in Campo Grande, Brazil. Following standard detections and identification, the authors are able to identify cohorts of Chlamydia and HPV positive cohorts of less than 20 cases. Next, using standard immune profiling assays, the authors are able to perform comparative assays. I have a few comments on the study

1. The authors do not provide any innovative reasoning behind the study design and its purpose. Previous studies on similar approaches have been published. While the authors claim that "Co-infection in the endocervical environment with other microorganisms such as Chlamydia trachomatis may increased risk of HPV infection and neoplastic progression.", the study fails to provide any conclusive evidence to this end? The authors need to state the hypothesis and the innovation involved

2. Chlamydial infections are highly prevalent in the young adolescents and sexually active age group whereas the data presented here is from women in a higher age range? The data hence may not be relatable to high incidence age cohorts and the effects of coinfections demonstrated in this study may not be reflective and of clinical significance.

3. Infections have been detected using PCR- how many of these cases had active v previous infection? immune presentation significantly changes in accordance with the status of infection? This data is required.

4. What are the cell types involved in cytokine signatures. Immune cell profiling data is required to be added to make better sense of the source of key cytokines.

5. While statistically significant, certain cytokine levels are marginally higher- how are these differences physiologically relevant? Additional corroborative evidence for the same is required to be added.

6. Minor- figure quality is very poor and needs to be enhanced.

6. PLOS authors have the option to publish the peer review history of their article (what does this mean?). If published, this will include your full peer review and any attached files.

Reviewer #1: **Yes: **Li Shen

Reviewer #2: **Yes: **Rishein Gupta

---

## [Author Response · Author response to Decision Letter 0]

16 Sep 2022

Response to Review’s Comments

(Manuscript Number: PONE-D-22-04538

"Cytokine profiling of samples positive for Chlamydia trachomatis and Human papillomavirus" (Original Article)

Dear Editor,

We appreciated all suggestions from reviewers about our manuscript. Please find below the answers to their specific comments/suggestions/queries.

Response to Journal requirements

1. Please ensure that your manuscript meets PLOS ONE's style requirements, including those for file naming. The PLOS ONE style templates can be found at https://journals.plos.org/plosone/s/file?id=wjVg/PLOSOne_formatting_sample_main_body.pdf and https://journals.plos.org/plosone/s/file?id=ba62/PLOSOne_formatting_sample_title_authors_affiliations.pdf.

Response: The manuscript was reviewed.

a) Did participants provide their written or verbal informed consent to participate in this study?

Response: We rewrite to stay more clear. “All patients who agreed to participate in this study signed a written Informed Consent Form. The study was approved by the Research Ethics Committee of the Federal University of Mato Grosso do Sul and the Fundação PIO XII under protocol no. 1468457 and 1635895, respectively.” (lines 129-133)

Response: The ‘Funding Information’ is correct. We remove the financial support in the manuscript. 

“This work was supported by the Federal University of Mato Grosso do Sul Foundation - UFMS/MEC - Brazil, the National Council for Scientific and Technological Development (CNPq) (Grant number: 406148/2016-3) and the Coordination for the Improvement of Higher Education Personnel (CAPES). This study was financed in part by the Coordenação de Aperfeiçoamento de Pessoal de Nível Superior - Brasil (CAPES) - Finance Code 001”

Response: We added the information suggested.

“Federal University of Mato Grosso do Sul (UFMS), Ministry of Education and Science (MEC), and the Hospital de Amor, Campo Grande-MS unit.”

“This work was supported by the Federal University of Mato Grosso do Sul Foundation - UFMS/MEC - Brazil, the National Council for Scientific and Technological Development (CNPq) (Grant number: 406148/2016-3) and the Coordination for the Improvement of Higher Education Personnel (CAPES). This study was financed in part by the Coordenação de Aperfeiçoamento de Pessoal de Nível Superior - Brasil (CAPES) - Finance Code 001”

Response: The ‘Funding Information’ is correct. We remove the financial support in the manuscript.

Response to Reviewer #1 Comments

Major Concerns：

Methods:

1. Patients: What is the age range of the individuals?

Response: The range of age is similar in all groups. The range of age was included in table 1 (Line 239).

2. What is the rationale behind the PCR testing HPV together with the human β-globin gene?

Response: Studies have widely used the Multiplex PCR with co-amplification of PGMY09/11 primers to detect HPV DNA and GH20/PCO4 for B-globin (as an internal reaction control) (listed below). This approach allows, in a single step, to carry out amplification of HPV DNA and verification of the quality of the genomic material, optimizing the time for diagnosis, resulting in reagent savings (detection of the PCR product) without losing the quality of the results.

1 - PLoS One 2021 Mar 22;16(3):e0248639. DOI: 10.1371/journal.pone.0248639. eCollection 2021; 

2 - J Med Virol . 2014 Feb;86(2):266-71. DOI: 10.1002/jmv.23725. Epub 2013 Sep 5

3 - BMC Pregnancy Childbirth. 2006 Sep 4;6:28. doi: 10.1186/1471-2393-6-28; 

4 - Gravitt PE, Peyton CL, Apple RJ, Wheeler CM: Genotyping of 27 human papillomavirus types by using L1 consensus PCR products by a single-hybridization, reverse line blot detection method. J Clin Microbiol. 1998, 36 (10): 3020-3027; 

5 - Gravitt PE, Peyton CL, Alessi TQ, Wheeler CM, Coutlee F, Hildesheim A, Schiffman MH, Scott DR, Apple RJ: Improved amplification of genital human papillomaviruses. J Clin Microbiol. 2000, 38 (1): 357-361.; 

6 - Fuessel Haws AL, He Q, Rady PL, Zhang L, Grady J, Hughes TK, Stisser K, Konig R, Tyring SK: Nested PCR with the PGMY09/11 and GP5(+)/6(+) primer sets improves detection of HPV DNA in cervical samples. J Virol Methods. 2004, 122 (1): 87-93. 10.1016/j.jviromet.2004.08.007).

3. Lines 166 and 538: a wrong reference (23) for Ct primers and detection..

Response: The reference 23 was changed.

Results:

1. If could be nice to include the cytokine levels from the individuals that had co-infection of Ct and HPV.

Response: “Due to the low number of co-infection samples, it was impossible to include the HPV x Chlamydia trachomatis co-infection in analyses (3/25) (S1 Table)” (Lines 235-237).

2. Lines 267-268: the tittle was unclear here. Perhaps it should be “comparison of cytokine levels in samples positive for C. trachomatis DNA with those in HPV DNA-positive samples”.

Response: The title was rewritten as suggested (Lines 259-261).

3. Table 2: column 4 “C. tracomatis negative” should be healthy control or no infections?

Response: The words “tracomatis negative” were changed to “healthy patients”.

4. If could be interesting to compare whether the cytokine levels with the individuals that had HPV DNA were different from the healthy controls.

Response: A comparison of cytokine levels of HPV-positive patients and healthy controls with these same samples was published in PLoS One. 2021 Mar 22;16(3):e0248639. DOI: 10.1371/journal.pone.0248639. eCollection 2021.

5. Figures 1 and 2 were blurred.

Response: The figures was enhaced.

 Minor Concerns：

1. Lines 77-78: “the elementary body (EB) that is extracellular, metabolically inactive…” this is not true. Recent evidence suggest that EB has low metabolic ability. Omsland A, Sixt BS, Horn M, Hackstadt T.FEMS Microbiol Rev. 2014 Jul;38(4):779-801. doi: 10.1111/1574-6976.12059. Epub 2014 Feb 24.

Response: This information was excluded (Lines 80-82).

2. Line 92: use a reference with women study is better.

Response: We changed the reference 6.

Response to Reviewer #2 Comments

1. The authors do not provide any innovative reasoning behind the study design and its purpose. Previous studies on similar approaches have been published. While the authors claim that "Co-infection in the endocervical environment with other microorganisms such as Chlamydia trachomatis may increased risk of HPV infection and neoplastic progression.", the study fails to provide any conclusive evidence to this end? The authors need to state the hypothesis and the innovation involved

Response: This study is not intended to be definitive or conclusive on the subject, but contribute to understand more about the interaction between the immune response and these two microorganisms, if the cervical or systemic microenvironment can somehow influence viral persistence and if the presence of C. Trachomatis can modulate this environment. We added this information in the introduction section (Lines 117-121).

2. Chlamydial infections are highly prevalent in the young adolescents and sexually active age group whereas the data presented here is from women in a higher age range? The data hence may not be relatable to high incidence age cohorts and the effects of coinfections demonstrated in this study may not be reflective and of clinical significance.

Response: The age group ranged from 26 to 64 years, with a mean of 44 years. 41,7% of patients aged between 25 and 40 years and 58,3% aged >40 and ≤64 years. The age group of patients included in our study is 25 years or older, as they are seen in health services in Brazil for cervical cancer screening, being submitted to Pap smear and detection of HPV when necessary. According to Beydoun et al., (2010) the prevalence under 25 years is slightly higher, and the differences are influenced by sociodemographic and behavioral factors.

J Women's Health (Larchmt). 2010 Dec;19(12):2183-90. doi: 10.1089/jwh.2010.1975. Epub 2010 Oct 15.

3. Infections have been detected using PCR- how many of these cases had active v previous infection? immune presentation significantly changes in accordance with the status of infection? This data is required.

Response: The existence of active or previous infection was not clinically investigated, the presence of CT was verified by the PCR technique, which is considered the gold standard for the diagnosis of this microorganism. When considering the immune response, the presence of this antigen in the cervical tract inevitably culminates in the formation of a specific immune response and production of cytokines in the cervical microenvironment, which can be disseminated systemically. This study is not intended to be definitive or conclusive in the clinic , therefore the aim of the work was to report the differences in the production of cytokines considering the presence of CT.

1)Expert Rev Mol Diagn. 2018 Aug;18(8):739-747. doi: 10.1080/14737159.2018.1498785. Epub 2018 Jul 19.

2)Best Pract Res Clin Obstet Gynaecol. 2002 Dec;16(6):789-99. doi: 10.1053/beog.2002.0322.

3) Clin Vaccine Immunol. 2017 Oct 5;24(10):e00203-17. doi: 10.1128/CVI.00203-17. Print 2017 Oct

4. What are the cell types involved in cytokine signatures. Immune cell profiling data is required to be added to make better sense of the source of key cytokines.

Response: The samples collected from the cervix and blood were not submitted for immunophenotyping but cytokine measurement using the Cytokine Bead Array (CBA) Th1/Th2/Th17 (BD Biosciences, San Jose, CA, USA). The purpose of the work was to consider the cytokines produced by the different groups. Future studies by our group are already considering immunophenotyping associated with cytokine dosage to characterize better the patterns obtained.

5. While statistically significant, certain cytokine levels are marginally higher- how are these differences physiologically relevant? Additional corroborative evidence for the same is required to be added.

Response: Regarding comparing IL-10 and INF-y concentrations in peripheral blood between positive and negative groups for DNA CT, we still observed significant differences in the DNA CT+ group, even with slightly higher levels. The same was observed concerning IL-4 and IL-2 in the ECC among the DNA CT + and DNA CT - groups. Regarding the cytokines IL-6 and IL-17, the results were also statistically significant and in this case the concentrations detected were much higher in DNA CT + patients (levels 50 to 150% higher). We know that the present work, being of a transversal nature, portrays a momentary panel of the systemic immune response and the uterine cervix. However, considering that the immune response occurs over a longer period, the high expression of a continued cytokine, even at slightly higher concentrations, may be sufficient to modulate the immune response profile. Thus, we can infer that the persistence of bacterial infection, due to lack of diagnosis or treatment, can cause an increase in cytokine expression for a sufficient time to modulate the immune response to cellular profiles that do not favor viral elimination.

6. Minor- figure quality is very poor and needs to be enhanced.

Response: The figures was enhaced.

---

## [Decision Letter · Decision Letter 1]

17 Oct 2022

PONE-D-22-04538R1Cytokine profiling of samples positive for Chlamydia trachomatis and Human papillomavirusPLOS ONE

Dear Dr. Puga,

Thank you for submitting your manuscript to PLOS ONE. After careful consideration, we feel that it has merit but does not fully meet PLOS ONE’s publication criteria as it currently stands. Therefore, we invite you to submit a revised version of the manuscript that addresses the points raised during the review process. Editors note: Please include a short paragraph to address the reviewer's remaining concern.

We look forward to receiving your revised manuscript.

Kind regards,

Thomas Forsthuber

Academic Editor

PLOS ONE

Journal Requirements:

Additional Editor Comments:

The authors have revised the manuscript according to the reviewers comments and satisfactorily addressed most of the comments.

Reviewers' comments:

Reviewer's Responses to Questions

**Comments to the Author**

1. If the authors have adequately addressed your comments raised in a previous round of review and you feel that this manuscript is now acceptable for publication, you may indicate that here to bypass the “Comments to the Author” section, enter your conflict of interest statement in the “Confidential to Editor” section, and submit your "Accept" recommendation.

Reviewer #2: All comments have been addressed

2. Is the manuscript technically sound, and do the data support the conclusions?

Reviewer #2: Partly

3. Has the statistical analysis been performed appropriately and rigorously? 

Reviewer #2: Yes

4. Have the authors made all data underlying the findings in their manuscript fully available?

Reviewer #2: Yes

5. Is the manuscript presented in an intelligible fashion and written in standard English?

Reviewer #2: Yes

6. Review Comments to the Author

Reviewer #2: The authors have made the best attempts to address comments raised by reviewers and have satisfactorily provided responses. However, given the nature of the study design and the limitations therein as indicated by the authors in the revision, it is advised to include a paragraph in the discussion section highlighting study design related caveats and the endeavor to undertake future/ ongoing studies to address them. This will serve as a good start point for other research groups investigation similar pathogens or study designs.

7. PLOS authors have the option to publish the peer review history of their article (what does this mean?). If published, this will include your full peer review and any attached files.

Reviewer #2: **Yes: **Rishein Gupta

---

## [Author Response · Author response to Decision Letter 1]

1 Dec 2022

Response to Review’s Comments

(Manuscript Number: PONE-D-22-04538

"Cytokine profiling of samples positive for Chlamydia trachomatis and Human papillomavirus" (Original Article)

Dear Editor,

We appreciated the suggestion from reviewer about our manuscript. Please find below the answer to his specific comments/suggestions/queries.

Response to Reviewer #2 Comment

1. The authors have made the best attempts to address comments raised by reviewers and have satisfactorily provided responses. However, given the nature of the study design and the limitations therein as indicated by the authors in the revision, it is advised to include a paragraph in the discussion section highlighting study design related caveats and the endeavor to undertake future/ ongoing studies to address them. This will serve as a good start point for other research groups investigation similar pathogens or study designs.

Response: “This study is not intended to be definitive or conclusive on the subject. Still, it contributes to understanding more about the interaction between the immune response and these two microorganisms, if the cervical or systemic microenvironment can somehow influence viral persistence and if the presence of C. Trachomatis can modulate this environment. The work aimed to consider the cytokines produced by the different groups with samples collected from the cervix and blood. However, studies by our group associated immunophenotyping with cytokine dosage to characterize better the patterns obtained. One of these studies is already published [50].” We added this information in the discussion section (Lines 421-429).

---

## [Editor Report · Decision Letter 2]

6 Dec 2022

Cytokine profiling of samples positive for Chlamydia trachomatis and Human papillomavirus

PONE-D-22-04538R2

Dear Dr. Puga,

We’re pleased to inform you that your manuscript has been judged scientifically suitable for publication and will be formally accepted for publication once it meets all outstanding technical requirements.

Kind regards,

Thomas Forsthuber

Academic Editor

PLOS ONE
---

## [Editor Report · Acceptance letter]

12 Dec 2022

PONE-D-22-04538R2 

Cytokine profiling of samples positive for *Chlamydia trachomatis* and Human papillomavirus 

Dear Dr. Puga:

I'm pleased to inform you that your manuscript has been deemed suitable for publication in PLOS ONE. Congratulations! Your manuscript is now with our production department. 

Kind regards, 

on behalf of

Dr. Thomas Forsthuber 

Academic Editor

PLOS ONE